# A Special Class of Experience: Positive Affect Evoked by Music and the Arts

**DOI:** 10.3390/ijerph19084735

**Published:** 2022-04-14

**Authors:** Emery Schubert

**Affiliations:** Empirical Musicology Laboratory, University of New South Wales, Sydney, NSW 2052, Australia; e.schubert@unsw.edu.au; Tel.: +61-2-93856808

**Keywords:** aesthetic emotion, self-actualisation, the arts, negative emotion, affective words, awe, being moved, positive psychology, coarse and refined emotions, qualia

## Abstract

A positive experience in response to a piece of music or a work of art (hence ‘music/art’) has been linked to health and wellbeing outcomes but can often be reported as indescribable (ineffable), creating challenges for research. What do these positive experiences feel like, beyond ‘positive’? How are loved works that evoke profoundly *negative* emotions explained? To address these questions, two simultaneously occurring classes of experience are proposed: the ‘emotion class’ of experience (ECE) and the positive ‘affect class’ of experience (PACE). ECE consists of conventional, discrete, and communicable emotions with a reasonably well-established lexicon. PACE relates to a more private world of prototypical aesthetic emotions and experiences investigated in positive psychology. After a review of the literature, this paper proposes that PACE consists of physical correlates (tears, racing heart…) and varied amounts of ‘hedonic tone’ (HT), which range from shallow, personal leanings (preference, liking, attraction, etc.) to deep ones that include awe, being-moved, thrills, and wonder. PACE is a separate, simultaneously activated class of experience to ECE. The approach resolves long-standing debates about powerful, positive experiences taking place during negative emotion evocation by music/art. A list of possible terms for describing PACE is proposed.

## 1. Introduction

Music and art (hence music/art) have been linked with wellbeing (to be read ‘improved/maintained wellbeing’) through numerous medically and psychologically grounded studies. Music/art appears to improve or maintain wellbeing through the positive valence emotions (e.g., happiness, joy, and calm) they evoke [1] and pp. 21, 29 in [2], [3,4]. However, it is not clear exactly what is meant by a positive emotion when experienced in response to music/art. Does it equate to pleasure (enjoyment, attraction, liking, etc.), and so the pleasure (or dispelling displeasure) itself is the conduit to wellbeing caused by music/art? Furthermore, the supposed evocation of positive valence emotions by music/art suggests that negative (valence) emotions cannot or should not be evoked if wellbeing is a desired outcome. Yet, there is overwhelming evidence that people can derive much pleasure from music/art evoking *negative* valence emotions (such as sadness, despair, grief, tension, etc.) e.g., [5,6,7].

Much important work has been published on the function, structure, and etiology of emotions evoked by music/art e.g., [8,9,10,11]. The present enquiry will focus, instead, on a different aspect of emotion in music/art, concerned with the experience itself. After all, understanding the phenomenology of how music/art feels, and finding ways to verbalise this, is paramount to building a more complete picture of the impact of music/art and for seeking nuanced and sophisticated connections between the subtleties of the experience itself and wellbeing outcomes.

Some have proposed that negative emotions experienced in response to music/art can occur because the negative portion of the experience is not in itself very important [12], because it is a means to an end (e.g., mediated by something that is more intrinsically positive) [8,13,14,15], or because working through the negative experience has a therapeutic, psychically cleansing impact that leads to the positive wellbeing outcome [16,17,18,19,20]. Another school of thought is that negative emotions make an invaluable and powerful contribution to the experience of music/art, with explanations suggesting that they trigger an intensity of emotion [21]; they have intertwined components (both positive and negative) built into them [22]; or that they operate in tandem with other, related affects (rather than being subsumed or subservient to them) [23]. The latter ‘co-existence’ of negative and positive emotion theories are more difficult to evaluate because they require one to demonstrate the simultaneous coexistence of apparently contradictory negative and positive emotions. However, such a solution would be highly parsimonious to the matter at hand.

Co-existence theories suggest that two kinds of qualitatively diverse experiences can occur at the same time and each contribute to the overall positive experience of music/art. Along that line of thinking, some pertinent solutions have been offered. One builds on complementary concepts referred to as emotion-valence (e.g., sadness) and affect-valence (e.g., being moved) [24], where affect-valence is related to the metaphorical temperature, charge or force/energy of the emotion-valence (the author of [6] traces through historical precedents for such metaphors). Another was proposed by Russell and Barrett [25] and consisted of a distinction between prototypical emotional episodes, which are discrete, definable, and usually directed at something (e.g., happy, sad, angry …) ‘what most people consider the clearest cases of emotion’ (p. 806), versus ‘core affect’, which refer to:

the most elementary consciously accessible affective feelings (and their neurophysiological counterparts) that need not be directed at anything. Examples include a sense of pleasure or displeasure, tension or relaxation, and depression or elation. Core affect ebbs and flows over the course of time. Although core affect is not necessarily consciously directed at anything—it can be free-floating (p. 806).

This distinction is reasonably consistent with Damasio’s [26] emotion and ‘feelings’ (the latter being aligned with affect). ‘Emotion’ (from these perspectives) is a reasonably well-established terminology that allows people to communicate their feeling states to others fairly reliably through a set of prototypical emotion words. However, much less understood is how affect valence/core affect is experienced and communicated. Using the same conceptual distinction (emotion-valence and affect-valence), the Affect Space Framework [27] proposed that experience of music/art take place when an object or event is perceived as being (usually) beautiful or sublime, and evokes in the perceiver ‘positive affect valence’ (whether accompanied by negative *emotion* valence or not). It is this positive affect valence that has been understudied, yet it is critical if we wish to better understand the nuance among positive experiences that occur through engagement with music/art and its consequent wellbeing benefits.

To make it clearer that the present paper is concerned with the *experience* of music/art—what it feels like [28,29,30]—I will refer to emotion-valence/prototypical emotion episodes (or just plain ‘emotion’) and affect-valence/core affect (or just ‘affect’) as two distinct *classes of experience*, and state that both classes can be experienced simultaneously but are usually fused, and hence are termed ‘emotion class of experience’ and ‘affect class of experience’, respectively. Since vocabulary for emotion class experience is established, this paper investigates the concept of affect class experience. To achieve this, a review of the literature is presented that focusses on research that does or could be conceived of as supporting the distinction in classes of experience, as well as offering nuanced descriptions of the positive affect class of experience that could be used to build an ‘affect class experience’ lexicon.

The review will commence with an overview of so-called ‘aesthetic emotions’, though these emotions alone, I will demonstrate, are untenable for the current purpose. I will then examine trends further back in time, commencing from the highly insightful observations by William James from the late nineteenth century through to more recent thinking in psychology as well as from the burgeoning subdiscipline of ‘positive psychology’. The ultimate aim is to produce an initial list of terms apt for describing the positive affect class of experience and in so doing provide a sense of the range of experiences that form part of this poorly understand class that is thought to be restricted to private, general, and more-or-less indistinct positive feelings.

## 2. Aesthetic Emotion Words

Aesthetic emotions are those emotions evoked by objects or events defined as having aesthetic value (usually as a result of being perceived as sublime or beautiful), and are an important, highly enriching part of human life. They are reported in response to sunsets, pieces of music, paintings, architecture, sport, and potentially any object or event. Experiences of awe, wonder, thrills, and being-moved are typical examples of aesthetic emotions, and these are particularly interesting examples because, apart from thrills, none of them are clearly and completely positive in valence but are, overall, positive, powerful experiences, each with a different nuance. Given the considerable attention paid to aesthetic emotions by philosophers; psychologists; and, more recently, neuroscientists, aesthetic emotions may provide a solution to the question of how to provide nuanced descriptions of positive affect experience evoked by music/art.

The modern, English-language conception of aesthetic experience has its roots in Western European thought from the Renaissance, with the introduction of the ‘aesthetic’ label attributed later to Alexander Baumgarten in a treatise dated 1735 [31]. Since then, theorising about aesthetics has occupied a significant literature in Western thought [32]. The expression ‘aesthetic emotion’ is even more recent, not receiving regular usage until the 20th century, and has come to refer to the central, visceral sensation of the aesthetic experience. While there is little consensus on the precise meaning of ‘aesthetic emotion’, the term has received sufficient attention to warrant consideration.

The set of aesthetic emotions as a lexicon for positive experiences in music/art show promise but also two considerable limitations. One limitation is that the concept has defied anything resembling a well-settled vocabulary. The emotions that are aesthetic are a matter of debate. They range from all emotions that are produced in response to an aesthetic object/event, e.g., [33], through to an exclusive subset that only particularly special aesthetic objects/events evoke [34] (for more detailed discussion, see [33]). To exemplify, in a comprehensive investigation of aesthetic emotions, Schindler et al. [35] proposed a 21 category (subscale) measure that constituted their Aesthetic Emotions Scale (‘Aesthemos’). They conveniently group the subscales (see Table 1), which allows us to ascertain how their usage is only partly suitable to the matter of interest here. In fact, in their classification system, the emotions that are commonly considered part of the special, smaller set are labelled ‘prototypical’ (see first row of the table), indicating that these subscales arise as a result of empirical usage rather than as a theoretical position (devised through research or introspection). Two of the other groupings (pleasing and epistemic emotions) could be incorporated into the present concept of positive affect experience; however, the final grouping in the table (negative emotion) does not because it refers to the adaptive function of negative experiences [36] (which in the present paper is related to the emotion class of experience).

This raises the second key limitation of aesthetic emotions. Those who consider intrinsically negative experiences such as disgust as aesthetic argue that there are positive outcomes from such engagement. However, these are usually related to moral, survival, self-reflective, or social change/improvement rather than necessarily being concerned with pure contemplation and the wellbeing that may arise from it, e.g., [37,38,39], or they are concerned with actual negative (unpleasant) experiences. The present paper investigated positive experiences because of their links to health and wellbeing, and so these kinds of negative (affect) experiences are not of concern here, but they add a considerable complication to applying aesthetic emotion to explain positive music/art experiences. The difference in the present endeavor of explaining positive experience evoked by music/art versus Aesthemos reflects the messiness and absence of sharp boundaries of aesthetic emotions, which Schindler, Hosoya, Menninghaus, Beermann, Wagner, Eid, and Scherer [35] acknowledge (p. 2), and is the challenge I am seeking to address through a review that now moves to more historically distant research in modern psychology.

## 3. Subtle, Coarse, Pseudo, and Real

An early example of two phenomenologically distinct classes of ‘emotion’ in modern psychology can be found in the writings of William James. He proposed that there are subtle emotions and coarse emotions. The coarse emotions are those that are explained by James’ famous theory of emotion generated through the primacy of bodily response (that is, an emotion is initiated by a bodily response)—when seeing an angry bear, I run, and so feel afraid, rather than running *because* I feel afraid. Regardless of the causal chain of events, these highly functional kinds of emotions align favourably with the emotion class of experience, and James also provided several specific examples of these: “grief, fear, rage, love, in which every one recognises a strong organic reverberation” [40] (p. 449).

The subtle emotions are aesthetic, and they are those that give pleasure through the form of the art work, its combinations of shapes, colours, aural properties, etc. This pleasure impacts so directly upon the individual that James refers to the pleasure as being of a primary kind. However, a ‘secondary pleasure’ also plays an important role, being linked to physical characteristics and feelings rather than particular emotions of the coarse variety. James explains:

These secondary emotions themselves are assuredly for the most part constituted of other incoming sensations aroused by the diffusive wave of reflex effects which the beautiful object sets up. A glow, a pang in the breast, a shudder, a fulness of the breathing, a flutter of the heart, a shiver down the back, a moistening of the eyes, a stirring in the hypogastrium, and a thousand unnamable symptoms besides, may be felt the moment the beauty *excites* us. ([40] (p. 470), italics as in the source)

The physical aspect of these responses resembles Sloboda’s [41] notion of using self-reported physical ‘emotion’ states, instead of emotion adjective vocabulary (emotion class experience descriptions). Sloboda argued that these physically based terms are well suited to describing music because they are memorable, distinct, and shared. The non-verbal nature of the physical symptoms that are associated with emotions were indicated as being pre or supra verbal: “They are arguably more closely connected to the experience of emotion than verbalizations which may be infected with rationalizations” [42] (p. 40), making the link between them and the concept of affect class experience plausible. Sloboda identified 12 physical emotions, with three of them reported reasonably frequently and consistently in response to music: tears (which included the symptoms crying and lump in the throat); shivers (consisting of goose pimples and shivers down the spine); and (least frequently) heart reactions (racing heart and pit-of stomach sensations).

James’ concept of primary and secondary subtle emotions also has a striking parallel with affect class experience, while the coarse emotions are commensurate with emotion class experience. Furthermore, the primary subtle emotions suggest a shallow hedonic tone (concerned with preference, liking, enjoyment, and so on [27]), and the secondary subtle emotions suggest deep hedonic tone (related to deeper, more intense and powerful positive experiences such as awe, being moved, and wonder, both part of the positive affect class of experience).

Another early example is in the monograph on the psychological basis of teaching by Thorndike [43]. He differentiates between ‘real’ emotions that serve the function of resting or invigorating body, mind, thought and action (somewhat in line with the emotion class) versus pseudo-emotions (a term used interchangeably with aesthetic emotions by Thorndike, p. 200), which gives ‘a very noble and specially to be desired form of pleasure” (p. 201). In these early examples, we already see clear distinctions that fit well with the proposed affect-emotion distinction, as well as the further subdivision of the affect class of experience.

Pratt, who critiqued James’ assertion regarding subtle emotions, nevertheless also made reference to a distinct class [44] (p. 175) of aesthetic emotions that included ‘feelings’ of ‘pleasantness, enjoyment, pleasure, rapture, elation, delight, transport, exultation, ecstasy’, which describe affect class experiences. He used the terms feeling and aesthetic emotions interchangeably and contrasted these with ‘real’ emotions (the same term used by Thorndike). He was convinced that there was a difference between enjoyment and pleasantness, but the explanation provided draws on attributing emotion to a work of art (such as a piece of music) rather than experience, leaving the individual free to enjoying its pleasantness:

How far can unpleasantness go before it is incompatible with aesthetic enjoyment? Can a person enjoy music and have an unpleasant emotion at the same time? Are there mixed feelings? Such troublesome questions as these must be answered if it is assumed that a person’s pleasure in a work of art can be accompanied by displeasure, but they need never be raised if it is discovered that the so-called emotions are really not emotions at all, but are characters of the music which bear a striking formal resemblance to emotion [44] (pp. 199–200).

Some philosophers refer to this as a cognitivist stance [45,46], with contradictions due to mixed feelings resolved by attributing the source of one emotion to the stimulus, and the felt experience being attributed to the other (the affect class). However, humans are able to, and do, experience these mixtures (but I suggest *classes*) of ‘feeling’ more or less simultaneously. The affect class of experience, which includes the subtle emotions for James, was lost by these and other researchers of the first half of the twentieth century (including James’ own later writing) as it became influenced by the emerging zeitgeist of behaviourism, which stifled thought on the matter, limiting emotions, feelings, and affects to purely observable behaviours [47].

## 4. Refined Emotions

Frijda and Sundararajan [48] developed the idea of refined emotions or ‘emotions of refinement’ as a concept highly compatible with the affect class and James’ subtle emotions. The concept ‘emotions of refinement’ was inspired by Chinese poetics and Confucian philosophy and drew on a broad range of research concerned with related concepts, thus marking a historical turning point in the amount of detail given to this additional affect class of experience. Refined emotion was differentiated from coarse emotions that align with the meaning James [40] uses, as well as the emotion class concept. Frijda and Sundararajan avoided listing sample emotions that could be considered exclusively refined, as well as an additional list that could be considered exclusively coarse. Rather, a wide range of emotions can be both coarse and refined. The categories, as with emotion and affect classes, overlap, but when entering the realm of refined emotion, we are focussing on the savouring of, and yet detachment from, the coarse emotion. They explain that refined emotions are not simply a subset of emotions but an elaboration of them:

[R]efinement represents a mode of perhaps all emotions that language or emotion taxonomy could distinguish. There exist refined anger, love, and sexual ecstasy, as well as coarse, straightforward anger, love, and sexual ecstasy. (p. 227)

Refined emotions are described by the authors as, for example, ‘noticing a felt bite or *glow*’, ‘*pondering* and *exploring* one’s misfortunes, aversions, or grief’, and they suggest that ‘[a]ction is absent, except for *contemplation* and *acceptance wriggles*’. The italicised terms (added) are possible affect class expressions that embellish the simultaneously experienced (coarse) emotion class. Refined emotion is therefore the phenomenological manifestation of emotional charge, heat, or force [6], being a separate class (or ‘mode’, using Fridja and Sundararajan’s term) while interacting with the emotion class of the experience.

Since the work of Frijda and Sundararajan, a burgeoning of classifications of affect and emotionally compatible distinctions have been published. Menninghaus et al. [49], for example, posit that psychological distancing acts as one of two mechanisms that furnishes the perceiver with safety and control over the aesthetic experience, an idea that has origins in ‘psychic distancing’ proposed by Bullough [50] and that resembles Frijda and Sundararajan’s concept of detachment. The second mechanism is enabled by the first and generates pleasure/enjoyment outcome for the perceiver. However, few studies have engaged with what I refer to as the affect class of experience, with the depth and compatibility of Frijda and Sundararajan’s refined emotions concept.

## 5. Flow, Absorption, and Concepts in Positive Psychology

In the psychological literature, there exists reasonably well understood experiences that can be seen as emotion-like while at the same time being quite distinct from emotion. The experience of flow, or being ‘in-the-zone’, is a period of intense, highly focused performance, where perception of time and space are altered through a highly motivated, deep level of absorption in the performance task itself. The experience is dependent on the level of skill the performer possesses and the level of challenge the performance task presents. High levels of both challenge and skill will produce flow experiences more readily than low levels of both, or an imbalance between the two. The experience of flow is therefore also apt to be considered an affect class rather than an emotion class of experience. Csikszentmihalyi [51] proposed a model of flow that sets it apart from a number of emotions. It is an absence of apathy, boredom, worry, and anxiety (ostensibly negative emotions) but is closer (in a semantic sense) to arousal and control, without being any of the emotion states in particular, suggesting that the sense of flow is itself a unique, emotion-like experience but phenomenologically distinct from emotion. It fits well with the present conceptualization of affect class experience because ‘the phenomenological experience of flow is a powerful motivating force’ [52] (p. 234).

Absorption is closely related to flow [53,54]. While flow refers to a state induced by engagement with an activity (production rather than just perception), the *experience* of flow is highly compatible with absorption during contemplative (not actively performing) music/art engagement. Indeed, absorption is quite likely an experiential component of flow [55,56]. Tellegen and Atkinson [57] refer to the honing of an individual’s attention when describing absorption: ‘’total’ attention, involving a full commitment of available perceptual, motoric, imaginative, and ideational resources to a unified representation of the attentional object’ [57] (p. 274). To demonstrate, the absorption scale devised by Tellegen has a rating item ‘When I listen to music, I can get so caught up in it that I don’t notice anything else’ [58], reflecting engagement with a thought or activity involving some loss of awareness outside that thought or activity. The same absorption scale also contains items that use vocabulary I have already argued is directly related to the affect class, such as the rating item ‘I can be deeply moved by a sunset’, suggesting that the construct of absorption, like flow, is an affect class of experience rather than emotion class.

The explicit link between absorption and affects but not absorption and emotion is further supported through the discovery of a more implicit relationships between absorption and the affect phenomena. Sandstrom and Russo [59] reported that ‘absorption may be an important moderator of the *strength* of emotional responses to music’ (p. 224, italics added), and, similarly, Kreutz, Ott, Teichmann, Osawa, and Vaitl [21] identified a positive relationship between emotional *intensity* (not specific emotions) and absorption. Kreutz and colleagues found that both intensely happy experiences and intensely sad experiences induced by music listening correlated with absorption, suggesting that the valence of the emotion (in this case whether happy or sad) does not exclusively influence the experience of absorption. Instead, it is the strength or intensity of an emotion that concurs with Charland’s [24] ideas of emotional heat, charge, or force. It is as though an emotion class of experience consists of a special component that allows the affect class of experience to percolate forth under the appropriate (e.g., typical music/art event) contextual conditions. This interpretation is similar to Tomkin’s [60,61] motivational amplifier theory where emotions drive or attenuate motivation, with motivation here being closely aligned to the affect class of the experience. Flow and absorption are powerfully positive experiences (regardless of the nature of the emotion with which the experience may or may not be associated).

Flow and absorption, I am arguing, belong to the positive affect class of experiencing music/art and should therefore be included in a definitive vocabulary of affect class experience. The argument for how affect class is a distinctly different experiential qualia should now be clear through these examples, despite the diffusive nature of the experience (in comparison to emotion class experiences). Another useful resource for locating a vocabulary that could be classified as the class of positive affect can be found in the emerging subdiscipline of positive psychology. Bryant et al. [62] observed that positive experiences may be related to one other but could also be organised in conceptually coherent ways. They presented 21 such terms in seven conceptual triads (the nature of the triads need not concern us here):

(a) elation, gladness, and joy …; (b) awe …, wonder …, inspiration …; (c) mindfulness …, insight …, and self-transcendence …; (d) hope …, optimism …, and positive thinking …; (e) imagination …, anticipation …, and positive daydreaming …; (f) absorption …, flow …, and peak experience …; and (g) flourishing …, capitalizing …, and savoring …. (pp. 61–62)

Bryant et al. continue, ‘[t]his list is just the tip of the ever-growing conceptual iceberg. Within each conceptual triad, should we consider these constructs to be distinct from one another or not? And how distinct are they?’ (pp. 61–62). The list presented by Bryant et al. includes only a very small number of overtly emotion class terms—gladness and joy—and they warn researchers adopting positive psychology terms without caution, asking ‘How can researchers know whether the positive constructs they seek to measure and explore are really distinct enough from existing constructs to warrant investigation?’ (p. 62). In the current proposal, I suggest that at the highest classificatory level of a taxonomy of these positive psychology terms would resemble affect class experiences.

Table 2 lists all terms proposed as verbalisation of affect class experiences identified in the reviewed literature. How they might be further grouped is a matter for future research. One possibility is presented in a provisional organisation of such a taxonomy in Figure 1. Several terms self-evidently fall into fairly neat subgroups. Deep and shallow hedonic tone is reasonably easy to classify, and the presence of physical correlates or symptoms of emotion class terms also seem to be a way of not only grouping affect class experiences but also reminding us that the latter can be intricately tied to emotions, and simultaneously activated, while remaining a distinct kind of experience.

## 6. Discussion and Conclusions

Lexicons evolve according to cultural pressures. At different points in historical time, members of the lexicon come into use while others depart, while others undergo changes in meaning and usage. As the communication about an area of thought, perception, or action becomes richer and more refined, there is pressure to adapt vocabulary accordingly [69,70]. For example, when individuals are asked to interact with a finely distinguished but novel set of subcategories, they are willing to label the subcategories using a consistently applied list of presented nonsense words, allowing for clearer communication about these finer level distinctions. However, those who do not interact at the finer level have no need to learn the new terminology except in more abstract terms [71]. As another example, Winter et al. [72] demonstrated that in the West, vision, being the most dominant of sensory modalities, also has the widest and most refined vocabulary.

Emotion words themselves present an important case in point. Diller [73] showed how the terminology used to describe emotions, and the precise meanings of the words, could drift over time. Within this drift, Diller observed a general increase in the number of terms used, with a period of relentless growth commencing from the 1200s CE to the present. Today, one can see an analogous situation with affect class experiences. The broad category of ‘emotion’ has become so cumbersome that it is fracturing (and according to some, has or never was a single broad concept) into new conceptual categories. I have argued that one such category is the ‘affect class’ of experience.

This paper has advanced the argument that music/art evoke (at least) two classes of experience simultaneously. The label for each class of experience I have proposed is emotion (the conventional kinds, such as happiness, joy, calm, sadness, anger, but not pleasantness, enjoyment, liking, etc.) and affect. Vocabulary for affect is limited and underdeveloped, with some theorists arguing that affects, like feelings, are not always accessible to conscious introspection and as a result have no accessible vocabulary. Affect class can be described in terms of experiences that may or may not be expressible through words but are qualitatively and importantly distinct from the emotion class of experience. To aid the reader in understanding how the two classes can be distinguished, the key distinctions detailed in the previous sections are summarised in Table 3.

The need for the distinction is brought into stark necessity by consideration of the enjoyment of negative emotion in music/art and to explain wellbeing effects. The emotion–affect distinction, instead of dismissing or diminishing the role of negative emotion evoked by music/art, acknowledges what people report. Negative *emotion class* experience in response to music/art (e.g., a piece of music evoking grief, a painting evoking the devastation through its depiction of a tragedy) is integral to an experience that is also reported as strong (deeply) positive affect. If one loves a tragedy, the love is the affect class of experience, and the evoked sense of tragedy is the emotion class of experience. They concurrently exist and are usually fused with one another rather than one (e.g., the negative emotion) subtracting from the overall positive intensity of the experience. For music/art to be an aesthetic experience producing positive wellbeing outcomes, it must, in some way at least, be positive, but this does not exclude negative *emotions* from being part of that experience and indeed the negative emotion fuels the intensity of the affect experience it activates.

This research has attempted to demonstrate that the private, ineffable experiences of art/music contemplation are highly varied and yet conceptually coherent at some level and that we therefore need to adopt and adapt vocabularies to better communicate about them, beyond being positive, powerful, strong, or intense. Terms such as awe, being moved, thrilling, crying, rapture, absorption, and several dozen terms capture some of the nuances of affect class experience, possibly building on emotions but as a distinct experiential class. The list of words proposed is based on usage by other scholars, in an attempt to set the groundwork for building a lexicon, reflecting a linguistic imperative to better understand the nature of the experience people have in response to music/art. The terminology has terms that link not just to recovering or improving wellbeing but something more. The lexicon connects us to concepts of transcendence and self-actualisation, experiences that may enhance wellbeing, e.g., [82,83,84] and energizing one’s ‘life force’, what Mathes [85] (paraphrasing Maslow’s [86] concept of self-actualisation) described as:

experiences in which the individual transcends ordinary reality and perceives Being or ultimate reality. Peak experiences are typically of short duration and accompanied by positive affect. Peak experiences teach the individual that the universe is ultimately good or neutral, not evil; that the ultimate good is composed of Being-values such as truth, goodness, wholeness, beauty, dichotomy-transcendence, aliveness, uniqueness, perfection, necessity, completion, justice, order, simplicity, richness, effortlessness, playfulness, and self-sufficiency; and that opposites really do not exist. As a result of having had a peak experience the individual is usually changed so that he or she is more psychologically healthy. (p. 93)

In a first step toward breaking down the nuances within affect class, experiences were labelled deep or shallow in hedonic tone. A possible third subclass was also proposed—terms depicting physical correlates of affect class experience, which may be reasonably distinct from shallow and deep hedonic tone.

If the conceptualization proposed in this paper is adopted, then a program of research is required to assemble the lexicon used to describe affect class experience and move toward evaluating experiences that may be currently considered ineffable (not yet able to be described in words). The proposed initial list of such words, and the nature of their internal structure and inter-relationships, is in need of further investigation to allow emotion/affect researchers to build a comprehensive picture of the influence of this key aspect of music/art experiences on people.

The taxonomy suggested (Figure 1) is not without limitations. I have examined the small but strong undercurrent of important research that has demonstrated that the two (and possibly more) experiential classes have well established precedents. For example, confusion exists over vocabulary that appears in both classes (such as joy and elation). A simple solution is to treat the terms as an emotion class experience and not part of the affect class vocabulary because the affect class consist of something that is (metaphorically and psychologically) emitted by another class of experience (e.g., the intensity or charge of experience generated in addition to the prototypical aspects of emotion class experiences such as joy or elation). Such decisions will require reconciling and calibration of theory with the purported experience itself. Such limitations should provide further impetus for a more detailed program of investigation regarding the plausibility and utility of the proposed approach.

The impact of music/art has sometimes been characterised in terms of the positive emotions evoked in the listener. Positive emotions (defined as emotions with positive valence by [87]) are one of the pillars of health and wellbeing according to the frequently cited PERMA (Positive Emotions, Engagement, Relationships, Meaning and purpose, and Accomplishments) model [88]. However, in research on music/art, positive emotions provide a broad, even simplistic account of how music/art impact the individual and their wellbeing. The present study has (1) demonstrated that a more nuanced approach is needed to understanding positive emotion, with the proposal made here based on consideration of the positive affect experiential class; (2) argued that the approach proposed is well grounded in terms of evidence garnered from the historical sources and strong undercurrents of more recent thought, particularly in positive psychology research, and in terms of theoretical plausibility; and (3) has commenced identifying the nuances of positive affect class experiences with an initial list of positive affect vocabulary to deal with the impoverished state of its lexicon, in contrast to the better established vocabulary of emotion class experience.

## Figures and Tables

**Figure 1 ijerph-19-04735-f001:**
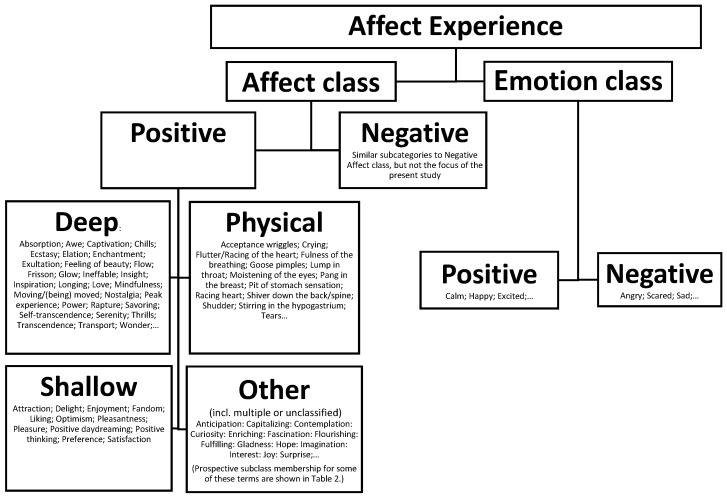
The provisional taxonomy of affect experience when contemplating music/art. See Table 2 for more details about each term sampled.

**Table 1 ijerph-19-04735-t001:** A grouping of 21 subscales in AESTHEMOS (Aesthetic Emotions Scale [35]) and their relevance to present study.

Grouping	Explanation of Grouping	Subscale Labels ^1^	Role in Present Study
Prototypical aesthetic emotions	“capture aesthetic appreciation irrespective of the pleasingness”	(1) feeling of beauty/liking, (2) fascination, (3) being moved, (4) awe (and, more weakly, (5) enchantment/wonder and (6) nostalgia/longing).	This links well to the proposed conceptualization of positive affect class.
Pleasing emotions ^†^	“all emotions with positive affective valence”	(7) joy, (8) humour, (9) vitality, (10) energy, and (11) relaxation	This links fairly well to the proposed conceptualization of positive affect class, but may also be well suited to the emotion class (e.g., relaxation).
Epistemic emotions *	“the search for and finding of meaning during aesthetic experiences”	(12) surprise, (13) interest, (14) intellectual challenge, and (15) insight	These subscales can be characterised as a positive affect class or as a separate experiential class.
Negative emotions	“emotions often are felt during aesthetic experiences that not only are unpleasant but also contribute to a negative evaluation regarding aesthetic merit”	(16) feeling of ugliness, (17) boredom, (18) confusion, (19) anger, (20) uneasiness, and (21) sadness.	Omitted because it could include an other-than-positive experience.

Note: ^1^ Subscale labels and explanations are taken from [35]. * Adopted as part of the affect class experience in the present paper; ^†^ adopted as part of the emotion class of experience as part of the present paper; other groupings not adopted.

**Table 2 ijerph-19-04735-t002:** The proposed list of vocabulary to describe positive affect class of experience.

Affect Term ^1^	Subclass ^2^	Source ^3^
Absorption	Deep	f
Acceptance wriggles	Physical	[48]
Anticipation	Shallow	e
Attraction ^PAE^	Shallow	[35]
Awe ^PAE^	Deep	b, [27,35]
Capitalizing		g
Captivation ^PAE^	Deep	[35]
Chills	Physical	[27]
Contemplation	Deep	[48]
Crying ^−^	Physical	[41]
Curiosity		z
Delight	Shallow	[44]
Ecstasy	Deep	[44]
Elation	Deep	a, [44]
Enchantment ^PAE^	Deep	[35]
Enjoyment	Shallow	[27,44]
Enriching	Deep	z
Exhilaration	Deep	[63]
Exultation	Deep	[44]
Fascination	Deep	z
Fandom	Shallow	z
Feeling of beauty ^PAE^	Deep	[35]
Flourishing	Deep	g
Flow	Deep	f
Flutter/Racing of the heart	Physical	[40,41]
Frisson	Physical	[27]
Fulfilling	Deep	z
Fulness of the breathing	Physical	[40]
Gladness *	Shallow	a
Glow	Deep	[40,48]
Goose pimples	Physical	[41]
Hope	Deep	d
Imagination	Shallow	e
Ineffable	Deep	z
Insight	Deep	c, [35]
Inspiration	Deep	b
Interest	Shallow	[35]
Joy *	Deep	a
Kama muta **	Deep	[63,64,65,66]
Liking ^PAE^	Shallow	[27,35]
Longing ^−,PAE^	Deep	[35]
Love *	Deep	[6]
Lump in throat ^−^	Physical	[41]
Mindfulness	Deep	c
Moistening of the eyes ^−^	Physical	[40]
Moving/(being) moved ^−,PAE,^**	Deep	[27,35]
Nostalgia ^PAE^	Deep	[35,67]
Optimism *	Shallow	d
Pang in the breast ^−^	Physical	[40]
Peak experience	Deep	f
Pit of stomach sensation	Physical	[41]
Pleasantness	Shallow	[27,44]
Pleasure	Shallow	[44]
Positive daydreaming	Shallow	e
Positive thinking	Shallow	d
Power	Deep	[67]
Preference	Shallow	z
Racing heart	Physical	[41]
Rapture	Deep	[44]
Satisfaction	Shallow	z
Savoring	Deep	g, [48]
Self-transcendence	Deep	c
Serenity *	Deep	z
Shiver down the back/spine	Physical	[40,41]
Shudder ^−^	Physical	[40]
Stirring in the hypogastrium ^†^	Physical	[40]
Surprise	Shallow	[35]
Tears ^−^	Physical	[41]
Thrills	Deep	[27]
Transcendence	Deep	[27]
Transport	Deep	[44]
Wonder ^PAE^	Deep	b, [35]

Note: ^1^ Some proposed affect class experience terms are accompanied by notes: ^†^ Hypogastrium (in the ‘Stirring in the hypogastrium’ entry) is the anatomical structure that best fits the ordinary language description ‘Pit of the stomach’ (see that entry). * Examples of terms that may be more commonly used to describe both emotion class and affect class experience. ^−^ despite being *positive* affect terms, these marked items have at least some potential negative connotations. It is the context (of contemplating or engaging with music/art) that enables these affects to be experienced as strongly positive. ** ‘Kama muta’ is an affect class related to being moved (by love) (see entry ‘Moving…’). It has also been linked to the physical experience subclass because the experience can include ‘tears, chills, warmth in the chest, feeling choked up’ (the latter represented by ’Lump in the throat’ in the table) [63]. Because the term is borrowed from the ancient Sanskrit language [68] and has been relatively recently proposed for adoption into English, it is currently not in common usage. ^PAE^ Classified as a ‘prototypical aesthetic emotion’ by [25]. ^2^ Proposed subclasses of affect class experience: hedonic tone (deep or shallow), and whether experience is described directly in terms of a physical correlate (physical). ^3^ ‘Source’ lists sample sources in which the term was located as an amendable member of the positive affect class of experience. A single letter denotation from a to g indicates one of the seven triads of the 21 positive psychology terms taken from Bryant, King, and Smart [62]. The single letter z is a term proposed in the present paper. Other sources cited in the table are (for more details, see References): [6]—Schubert (2013); [27]—Schubert, North, and Hargreaves (2016); [35]—Schindler et al. (2017); [40]—James (1890); [41]—Sloboda (1991); [44]—Pratt (1931); [48]—Frijda and Sundararajan (2007); [63]—Zickfeld et al. (2019); [64]—Fiske, Schubert, and Seibt (2017); [65]—Steinnes et al. (2019); [66]—Fiske, Seibt, and Schubert T. (2019); [67]—Schubert, Hargreaves, and North (2019).

**Table 3 ijerph-19-04735-t003:** A summary of conceptualisations distinguishing (positive) affect class and emotion class.

C	(Positive) Affect Class	Emotion Class	S
Structure	Refined, subtle emotions (James) OR a wide range of emotions can be both coarse and refined (Frijda and Sundararajan [48]). Different terms may have considerable overlap in meaning with other affect experiences and may be difficult to clearly distinguish from one another.	Coarse emotions. Generally easy to distinguish from one another.	[40]OR [48]
Sample ^1^	Awe, moved, wonder, thrills, absorbed, and energised.	Sad, angry, scared, calm, happy, and excited.	[25]
Feels like	Savouring of, and yet detachment from, the coarse emotion.	The coarse emotion itself, e.g., feeling sad or feeling happy.	[48]
Directedness	Diffuse; difficult to poinpoint the sensation to a particular, unique type, or to a particular object/event, apart from the object/event being engaged with (e.g., frisson, thrills, and chills are overlapping concepts and subtley distinguishable from one another [74,75,76]). The experience may be undescribable in words (ineffable).	Specific, identifiable, self-contained experience (e.g., I feel happy, I feel sad, etc.); focussed on the self or the object causing the emotion (i.e., not as diffuse).	[6,24,25,26]
Lexicon	The terminology is poorly established and needs to consider the ineffable. With the possible exception of some ‘aesthetic emotions’ (awe, being moved, wonder, and thrills), there is no prototypical terminology; core affect.	Terminology well established. Prototypical: discrete, definable.	[25]
Structure:	An intensity or strength of feeling eminating (usually) from the emotion. Metaphors with temperature (heat), charge, or force/energy of the coarse emotion; embellishment of simultaneous emotion class experience; wholly or in part positive.	Consists of physiological, signalling, feelingful, and motivational components; valence, arousal; can be positive or negative.	[6,21,22,24,48,77,78,79]
Function	Powerful, motivating force, without necessarily knowing why (apart from the act of engagement with the object/event); arises through pure contemplation/engagement with a thought, object, or event and is in and of itself a positive; supports exploration of emotions in a safe environment and possibly generates wellbeing.	Various: attraction, neutral, repulsion (including withdrawal, attack); knowing why (e.g., the cause/trigger of the feeling); adapting to the environment; moral, self-reflective, or social change/improvement.	[16,25,36,52,61,80,81,82,83,84]

Note: The purpose of the table is to aid the reader to digest the somewhat nebulous distinction between the two classes of experience (affect and emotion) in a music/art context (where affect is positive). To help the reader better understand the two concepts, non-experiential comparisons are shown, though some are clearly apt for also describing phenomenological distinctions. Note also that non-experiential aspects are neither necessary nor sufficient for the proposed distinction between the *experience* classes. It is also worth noting that motivation has been described as a component of emotion (see Structure row), while affect has a *function* (see that row) of motivational drive. That is, motivation is more central to affect, but the emotion component drives it. This kind of linking can be found, for example, in the theory of emotion as a motivational amplifier of Tomkins [60,61]. C = conceptual aspect.S = sources upon which interpretations of the conceptualization are based.^1^ See Table 2 for the more extensive, proposed list of affect terms.

## Data Availability

All data are presented in the body of the article and so no additional data availabiity statement is needed.

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
