# Peer review of "A Special Class of Experience: Positive Affect Evoked by Music and the Arts"

_ijerph, 2022, doi:10.3390/ijerph19084735_

Round 1

Reviewer 1 Report

The subject under investigation was presented in detail. What effects can be derived for the integration of music in medicine and what can be benefited from?

Author Response

Reviewer 1

Comments and

Suggestions for Authors

The subject under investigation was presented in detail. What effects can be derived for the integration of music in medicine and what can be benefited from?

Response: I thank the reviewer for the encouraging comments, and for the question of the effects and benefits.  In the final paragraph of section 6, I clarify the nascent state of our understanding of the positive class experience, and that more needs to be understood about it to support our understanding of how music in medicine is integrated from this perspective.  The benefits may be numerous—a matter for further research—the strongest evidence comes from the role of positive emotions in the PERMA model of health and wellbeing.  Specifically, -the final paragraph states “Positive emotions [defined as emotions with positive valence by 87] are one of the pillars of health and wellbeing according to the frequently cited PERMA (Positive Emotions, Engagement, Relationships, Meaning and purpose, and Accomplishments) model [89].  However, in research on music/arts positive emotions provide a broad, even simplistic account of how the music/arts impact on the individual and their wellbeing.” – Hence the potential benefits are very strong, but not yet well understood because we do not sufficiently understand the proposed (but as argued in the paper, critical) concepts of positive affect class experience.

Reviewer 2 Report

Thank you for the contribution of your scholarship to IJERHP. I found that your topic is valuable, and the writing style is appropriate. However, I think that this very broad topic was not adequately referenced. By combining responses to music and the arts, you are trying to connect responses to performing arts (music, dance, theater), which take place in real time, and visual arts (painting, sculpture), which are analyzed reflectively.

Philosophers from Aristotle to Immanuel Kant have had much to say on the affect evoked by music and the arts. Perhaps their contributions should be mentioned in the article. Quantitative researchers have also examined affective responses to music, but you have not referenced any of them (Kate Hevner in the 1930s, Albert Leblanc in the 1980s, John Kratus in 1993). There are also reviews of research on the impact of music on children and young people (Susan Hallam, 2010).

You cite Leonard Meyer in your reference list, but do not mention him in the article. Meyer wrote that the feelings elicited in music are neither positive nor negative, which would contradict the conclusion of your paper.

My suggestion is that you limit your discussion to one of the arts, rather than try to examine the effect of all of the arts.

Author Response

Reviewer 2

Comments and

Suggestions for Authors

Thank you for the contribution of your scholarship to IJERHP. I found that your topic is valuable, and the writing style is appropriate. However, I think that this very broad topic was not adequately referenced. By combining responses to music and the arts, you are trying to connect responses to performing arts (music, dance, theater), which take place in real time, and visual arts (painting, sculpture), which are analyzed reflectively.

Response: Agreed that several forms of art are being combined in this paper.  They are referred to under the umbrella of the arts, and it is the special status of the arts (not necessarily just music or just painting, or just theatre etc.), that poses the interesting question.  This is why reference is made to the ‘object or event’ – so as to encompass all art form outputs.  The proposed conceptualisation of positive affect class experience is self-evidently applicable to all art forms, regardless of their dependency on temporality or space (or both), and so the limitation seems unnecessary at this early stage of investigation.

Philosophers from Aristotle to Immanuel Kant have had much to say on the affect evoked by music and the arts. Perhaps their contributions should be mentioned in the article.

Response: Reference to these important philosophers have been added – Kant (as well as Shaftsbury and others) being particularly relevant to the development of the aesthetic emotion concept (see line 125 of revised ms – with an excellent discussion by Townsend cited), and Aristotle (see line 49 of revised ms) to explaining the pleasure of negative emotion.  Thank you for the suggestion.

Quantitative researchers have also examined affective responses to music, but you have not referenced any of them (Kate Hevner in the 1930s, Albert Leblanc in the 1980s, John Kratus in 1993). There are also reviews of research on the impact of music on children and young people (Susan Hallam, 2010).

Response:  Yes, agreed that these are indeed important researchers of emotion (in music, in particular).  However, these researchers, and the papers cited (with the exception of Hallam) are concerned with how musical features impact on a selection of emotions, rathe than the experience of art works.  It would be nice to add something about the influence of musical features in the manuscript, but I argue that the ms is already dealing with complex concepts related to definitions of emotion already, and the relationship between musical features and emotions is better dealt with elsewhere, rather than further adding to the strands in the paper.  The Hallam paper (presumably the seminal paper DOI: 10.1177/0255761410370658) reviews the non-musical benefits of music (in areas such as literacy, numeracy, intellectual development and so on), and so it is not of direct relevance to the topic under investigation.

You cite Leonard Meyer in your reference list, but do not mention him in the article. Meyer wrote that the feelings elicited in music are neither positive nor negative, which would contradict the conclusion of your paper.

Response: Meyer was mentioned in the third paragraph of the Introduction (section 1) it states “Some have proposed that negative emotions experienced in response to music/art can occur because … it is a means to an end (e.g., mediated by something that is more intrinsically positive) [8,13-15]” -  reference 15 is Meyer, (1956).  This is in reference to Meyer’s concept of generalised arousal in response to music, in line with R2’s observation. 

My suggestion is that you limit your discussion to one of the arts, rather than try to examine the effect of all of the arts.

Response:  Please see response made to R2’s first point.

Author Response

Reviewer 3

Comments and

Suggestions

Response: No specific comments or suggestions were made by R3, and their overall assessment was positive, for which I am grateful.

This manuscript is a resubmission of an earlier submission. The following is a list of the peer review reports and author responses from that submission.

Round 1

Reviewer 1 Report

It is a very interesting study which resolves the long standing debates about powerful, positive experiences taking place during negative emotion evocation by music/art.

Editing of spaces is required.

Reviewer 2 Report

**** beginning of review

NOTE: My background is in cognitive psychology and music cognition, and my review is written from this perspective. I am less familiar with research on emotion and affect. Nonetheless, given that the manuscript is presumably aimed (at least in part) at readers without expertise in emotion and affect, seeing my questions and confusions should help the author to prepare a draft that might be more accessible to those without that expertise.

General Issues:

Co-existence. The coexistence idea is interesting, but it isn’t clear when two elements might blend (e.g., color mixing, pointillism) or remain separate (e.g., individual notes of a musical chord). It was also a bit confusing as it wasn’t always clear which two elements might be considered as co-existent. In some cases, the co-existence seems to involve positive and negative valence (e.g., lines 52-53), but in other cases, the co-existence seems to involve emotion and affect (e.g., lines 89-91). Perhaps this can be clarified.

Value. As aesthetic emotions are evoked by objects or events having aesthetic value (lines 107-108), presumably non-aesthetic emotions are evoked by objects or events that do not have an aesthetic value. However, it isn’t clear what the nature of “value” is, and it would be helpful if that nature were clarified. Would the value of an “awe” be the same as the value of a “thrill” (line 111)? If an aesthetic value is necessary for an aesthetic emotion, then what can facilitate or inhibit arising of aesthetic value needs to also be discussed.

Flow. I didn’t understand the suggestion that flow was emotion-like (lines 278-291). The author states that high levels of challenge and skill produce flow and that flow is dependent upon the skill of the performer. Thus, flow (at least in most discussions) occurs during some sort of performance or production. However, an aesthetic emotion often arises in the absence of activity (unless passivity such as gazing at a sunset or listening to music is considered a performance or production of the observer), and aesthetic emotion (as described in the manuscript) arises from the activity of the perceiver and not from the activity of the performers. It would be helpful if a specific list of similarities and differences were provided. After laying out the reasons why flow was emotion-like, the author then concludes that flow is phenomenologically distinct from emotion, and so I’m not clear on why the flow notion was initially introduced.

Specific Issues:

Lines 38-40: Is it correct to say the present enquiry focuses on aspects of emotions? It seems like the present enquiry is focused more on trying to distinguish between emotion and affect and on positive elements of these two categories.

Lines 45-54: A seminal contribution that isn’t mentioned is that negative emotion (in the form of arousal or tension) that arises from a mismatch between what is expected and what is perceived can contribute to an aesthetic effect (e.g., Berlyne, 1971; Huron, 2006; Meyer, 1956).

Line 55: “more difficult to evidence” is nonstandard. Did the author mean “more difficult to evaluate”?

Lines 62-64: How does “metaphorical temperature, charge, or force/energy” relate to the more common dimension of arousal?

Lines 79-84: I was a bit surprised that the affect grid (e.g., Russell, Weiss, & Mendelsohn, 1989) wasn’t mentioned, as that framework (pleasant-unpleasant; aroused-unaroused) seems quite relevant and is commonly used.

Line 122: Aesthetics was a significant part of early psychophysics in the nineteenth century, as well (e.g., Fechner’s experimental aesthetics as described in his book Vorschule der Asthetik).

Line 127: “The set of aesthetic emotions as a lexicon for positive experience in music/art show promise…”. The suggestion seems to be that terms adapted from an understanding of aesthetic emotions can be used to describe positive experience, but wouldn’t that actually confuse rather than clarify matters? I can have positive experiences that aren’t necessarily aesthetic, yet if I applied a term used for aesthetic experience to those non-aesthetic positive experiences, then I would lose the precision of definition. In order to avoid confusion, shouldn’t the approach be to have separate terms for separate domains? Or is using aesthetic terms for non-aesthetic positive experiences assumed to be metaphorical?

Lines 136-139: I didn’t understand this sentence. Why might some group of emotions be thought to arise from a theoretical position rather than experience (or usage)?

Lines 139-143: So negative emotions are not positive affect because negative emotion is limited to negative experience? Isn’t that a bit obvious? Or is there some important subtlety that I’m missing?

Lines 152-156: I’m not convinced that positive outcomes from negative experiences are limited to moral, self-reflective or social change/improvement rather than wellbeing. For example, I might be hungry but disgusted at the sight or smell of spoiled meat, and that disgust can aid my wellbeing by forcing me not to eat the spoiled meat.

Lines 168-170: The author suggests that coarse emotions depend upon bodily responses, and by implication, subtle emotions do not depend on (or at least are less dependent on) bodily responses. However, wouldn’t some types of aesthetic emotions (which appear to be similar to subtle emotion) involve bodily responses (e.g., chills or shivers down the spine, etc.)? It isn’t clear why this difference is important.

Lines 190-193: This distinction isn’t clear. Wouldn’t (verbal) self-reports of emotional states necessarily involve an emotion adjective vocabulary? How might one provide such a (verbal) self-report without using such a vocabulary?

Line 202-214: By this point in the manuscript, I started thinking that readers could really use a table in which all the different distinctions (and equivalences) are clearly listed.

Line 222: The open parenthesis that should be paired with this close parenthesis appears to be missing.

Line 226: “can a person enjoy music and have an unpleasant emotion at the same time”?  Well, yes (as the author subsequently admits on line 236). Lots of people like scary or horror movies, which on the surface, at least, suggest the co-occurrence of enjoyment (positive) and fright/horror (negative) at the same time.

Lines 230-231: “characters of the music which bear a striking resemblance to emotion”. I know this is not the author’s claim but a citation of someone else, but I didn’t understand this at all. Music is, ultimately, a succession of notes in time. In what possible sense can that be said to resemble an emotion? Also, should “characters” be “characteristics”?

Line 262: If refined emotions involve noticing an action done to one’s self (e.g., a felt bite) but inaction on one’s part, then are refined emotions necessarily passive?

Line 304: I think “affecs” should be “affect”.

Lines 306-317: It seems like the fundamental difference between absorption and emotion is that emotion reflects the content of the experience and absorption reflects the intensity of the experience. If that isn’t correct, then perhaps this could be revised to make such an interpretation less likely.

Line 319: The author earlier suggested that flow was phenomenally distinct from emotion (line 291), but it isn’t clear why flow is thus (necessarily?) an affect.

Lines 348-349: What does it mean to say that positive psychology terms “resemble” affect class experiences? Taken literally, the author seems to suggest that positive psychology and positive affect are the same thing (the terms in each refer to the same underlying construct). Is that really the case, and if so, what is the justification? (I think that Table 2 might be intended to provide some of the answer, but the reader will need more explanation.) More specifically, does “resemblance” or “similarity” involve identity? Can this be elaborated?

Line 389: How can something evolve if there is stagnation? As evolution involves change, and stagnation involves a lack of change, these seem like opposing or contradictory notions.

Line 394: What does “refined” in “refined but novel” mean? Is this related to the idea of “refined” emotions mentioned earlier (lines 243-267)?

Line 417-419: This sentence is confusing; either some punctation or some words might be missing.

Line 424: When I think of the results of two processes “adding together”, I think the result is larger in magnitude if the two processes are operating in the same direction and smaller in magnitude if the processes are operating in opposite directions. Thus, if emotion and affect really “added together”, a negative emotion should cancel out the effects of a positive affect. So, perhaps rather than “added together”, the author might consider something like “concurrently exist and influence experience”. Or have I misunderstood?

Line 430: Does “is itself” describe the ideas in the paper of the construct to which those ideas are intended to refer? This is not clear.

Lines 435-436: Does the manuscript really attempt to “create a lexicon”? It seems to argue for the need of a such lexicon, but not actually create one. Or have I missed some subtlety?

Lines 438-440: This seems a little exaggerated. At the very least, a detailed example in which the reader is talked through the stages of such a connection is needed.

Line 459: Is the “vocabulary” referred to here the same as the “lexicon” referred to previously? If so, it would be easier on readers if a single term was consistently used. If not, can the difference be clarified?

Lines 460-461: There seems something inherently contradictory, or at least incomplete, in the notion of evaluating the words for an experience that cannot be deserved in words.

Line 465: Maybe I’m missing the obvious, but I’m still not entirely clear on what the taxonomy is. Is it merely affect – subclass – source? Perhaps a figure or diagram would help?

Line 477: There are many potential impacts of music and art. Although readers can probably figure out which impact the author is referring to, it would be better if that were explicitly stated.

**** end of review

Reviewer 3 Report

Thank you for your contribution to the International Journal of Environmental Research and Public Health.

In my view, the opening paragraph of a research article should invite the reader to explore the study and its research findings. Please re-read the opening two sentences of your article. “Music and the arts in general (hence music/arts) have been linked with wellbeing (to be read ‘improved/maintained wellbeing’) through numerous, medically and psychologically grounded studies. Music/arts appear to improve or maintain wellbeing through the positive valence emotions (e.g. happiness, joy, calm) they evoke [1å,2 p. 21 & p. 29,3,4].” I suggest that you rewrite this convoluted opening to be more reader-friendly.

Regarding the pleasure that listeners may derive “from music/arts evoking negative valence emotions” such as despair and grief, I suggest that you consider not just theories but evidence. Sloboda, for example, refers to shivers and heart reactions, but these are not the same as despair and grief. “Despair” is a complete loss or absence of hope. “Grief” is deep sorrow, especially that caused by someone's death. Why would a listener choose to engage with music that provides an absence of hope or a sorrow over someone’s death? I think that you are aiming too high in your expectation that listeners seek out negative emotions. There is a difference between music that evokes deep feelings and sounds that evoke an absence of hope. The reader should expect psychological evidence of your proposed

There is evidence connecting musical characteristics such as tempo, mode, and meter to emotional responses (Hevner, 1935, 1936, 1937; Kratus, 1993). This article cites none of it.

Csikszentmihalyi’s theory of flow relates to the challenge and skill of the performer, not the listener’s experience

I’m sorry but Table 2 is not very persuasive. Who are these sources? Who determined the subclass?

I suggest that you revise the article to provide more persuasive evidence for your hypothesis.
